# Low-Rank Networks Learn High-Frequency Features on Loss Landscape Valleys

## Abstract

We ask how the **geometry of the loss** (flat regions, sharp drops, narrow valleys) connects to **what the network learns** (which frequencies, which channels do what) in a simple setting: low-rank neural networks trained to fit sums of cosines in one dimension. We give three linked results. **(1) Channel specialization:** we formalize when one "channel" dominates at a given input (a log-ratio criterion) and show experimentally that learning rate and batch size control whether channels specialize or collapse. **(2) Oscillatory complexity:** we count how many peaks and valleys each channel's output has along the input; this count grows with depth, so deeper layers learn more wiggly, high-frequency structure. **(3) Loss shape and frequency:** the training loss stays flat for long stretches, then drops sharply; each flat stretch corresponds to a frequency not yet learned, and each drop corresponds to the network entering a narrow valley in the loss landscape. Reducing the learning rate is what allows the optimizer to enter these valleys. We also show that plateau-escape time scales with learning rate and batch size, and that the low-rank structure preserves the symmetry of the target. All experiments use a single, reproducible setup (1D regression, cosine targets, RF-LR architecture); we do not claim the same picture holds beyond this setting.

## 1 Introduction

We study how *low-rank random feature* (RF-LR) networks learn representations and how this ties to the loss landscape Chizat & Bach (2018); Abbe et al. (2023) in 1D regression on frequency-dependent (cos) targets.

Our contribution is a **unified picture** in three strands. **(i) Feature learning and the log-ratio:** channels specialize to distinct spatial locations and progressively capture higher frequencies; a mean-field toy model Nguyen & Pham (2023); Pham & Nguyen (2021) and a conditional *log-ratio criterion* characterize when one channel dominates; we validate with log-ratio trajectories and diagnostics (collapse vs. convergence with LR/batch). **(ii) Oscillatory complexity:** the mean number of strict local minima per channel (layer-wise partials) increases with layer index, consistent with deeper layers learning more oscillatory representations. **(iii) Valley–frequency correspondence:** the loss has plateau–sharp–plateau structure (saddle-to-saddle); each plateau is a frequency not yet learned; the landscape is a plateau with sharp, deep valleys reachable only when LR is small—at each LR divide the loss drops and partials refine; we report plateau-escape scaling with LR and batch and show that low-rank preserves target symmetry while full-rank can learn asymmetric features. Observables (mean minima per channel, $2L$-spike pattern per layer $L$) bridge landscape and feature learning. Scope: 1D cos functions/RF-LR; we do not claim generalization beyond this setting.

## 2 Model

We work with **low-rank random feature (RF-LR)** networks Zhang et al. (2025) on 1D regression with cosine targets. The architecture uses a tall-skinny bottleneck: the first-layer features $L^0$ and the mixing matrices $L^{(\ell)}$ are frozen, and only the channel weights $w_\ell$ are trained. Target, data, and the full training protocol are detailed in Appendix A.

## 2.1 METHODOLOGY

We use a single experimental protocol: sum-of-cosines targets on $[-1, 1]$, RF-LR networks with adaptive learning-rate schedule and checkpoints before and after each LR divide, and observables including loss, fit, channel partials, log-ratios, and mean minima per channel. Full config naming, baseline sweeps (ranks 5, 10, 20), plateau-escape and log-ratio runs, and exhaustive methodology are given in Appendix C.

## 3 RESULTS

When trained with SGD, the loss curve spends long stretches on flat plateaus before dropping sharply; each plateau corresponds to a frequency that the network has not yet captured, and the dynamics move from saddle to saddle Abbe et al. (2023). The loss landscape effectively presents a plateau riddled with narrow valleys: the optimizer wanders on the plateau until a learning-rate reduction allows it to slip into one of these valleys. When that happens, the loss drops and the fit improves, with high-frequency structure emerging in the middle of the input domain and in the channel partials (Figures 1, 2). As each new frequency is acquired, the partial functions become more oscillatory, and the mean number of strict local minima per channel goes up.

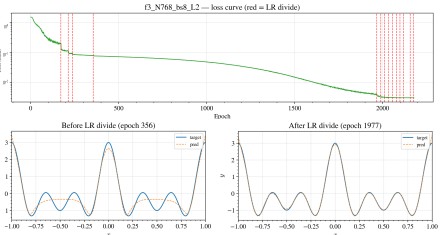 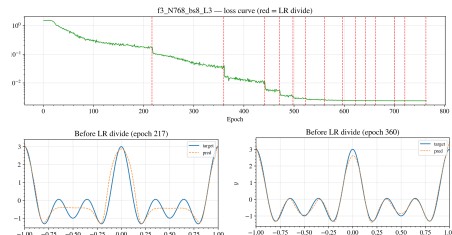

Figure 1: **Loss and fit around learning-rate reductions.** Left: depth $L = 2$; right: $L = 3$. Training on a sum of three cosines ($N = 768$, batch 8). The curves show how the global fit (blue) approaches the target (orange) in stages: after each LR cut the loss drops and the predictor gains finer oscillations in the interior of $[-1, 1]$, reflecting descent into a deeper valley of the landscape.

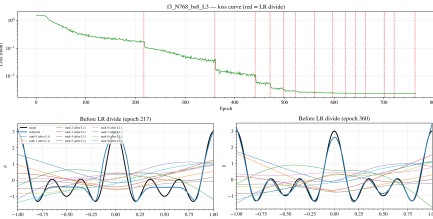

Figure 2: **Channel partials before and after LR divides** (depth $L = 3$, same setup). Each curve is the scalar output of one low-rank channel over the input domain. After long training and an LR reduction, the partials develop visible high-frequency structure in the middle of the interval, matching the moment when the trajectory enters a sharp valley and the network picks up higher modes.

We track how channels specialize using the log-ratios $R_{i,j} = \log|f_i| - \log|f_j|$ at a fixed point $x = 0$. When channels separate by role, the maximum of these ratios grows over training; when they collapse to a similar profile, the ratios stay flat or behave erratically. Figure 3 shows how the maximum log-ratio evolves and how the full set of pairwise ratios at $x = 0$ is distributed over training; a spread, roughly uniform distribution in log space indicates that different channels take on distinct roles.

We also count, per layer, how many strict local minima each channel partial has on a grid; this count rises with depth, so deeper layers carry more oscillatory structure. We conjecture that layer $L$ exhibits on the order of $2L$ spikes in the partials; the mean number of strict local minima per channel is a proxy for this oscillatory complexity. Figure 4 shows the mean number of strict local minima per channel versus layer index for a network with $L = 6$ blocks.

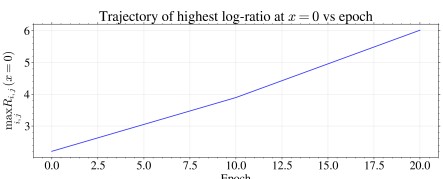 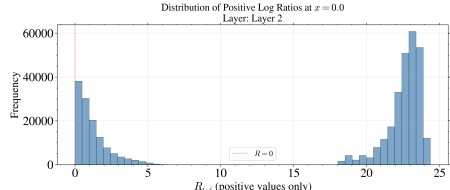

Figure 3: **Log-ratio evolution at** $x = 0$**.** Left: the maximum log-ratio between channel pairs grows over epochs when channels specialize. Right: the full set of pairwise log-ratios over training; a spread, roughly uniform distribution in log space indicates that different channels take on distinct roles.

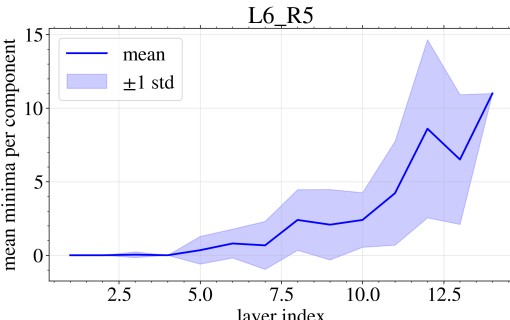

Figure 4: **Oscillatory complexity across layers.** Mean number of strict local minima per channel (on layer-$\ell$ partials) vs. layer index $\ell$ for a network with $L = 6$ blocks, width 128, rank 5. The rise with depth is consistent with the conjecture that layer $L$ exhibits on the order of $2L$ spikes and that later layers develop more oscillatory internal representations.

On symmetric targets such as sums of cosines, full-rank MLPs often learn asymmetric internal representations, while the low-rank setup tends to preserve symmetry in the channel partials (Figure 5). The time to escape the first plateau scales roughly inversely with learning rate, and smaller batches (e.g. 1–4) escape more readily (Figure 6). Deeper networks can exhibit on the order of $2L$ spikes per layer in the partials; very deep nets sometimes become unstable.

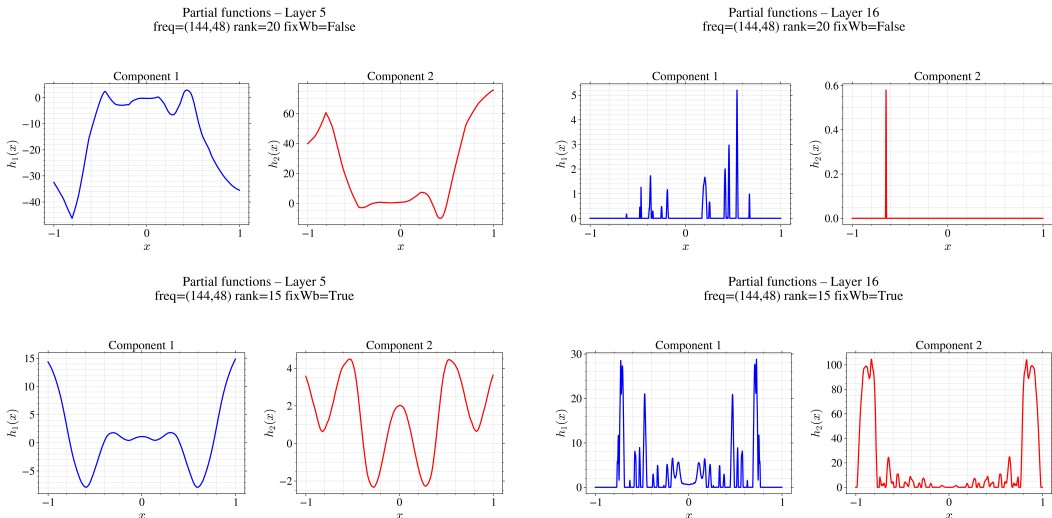

Figure 5: **Symmetry preservation: low-rank vs. full-rank.** Top row: full-rank MLP (random features removed) at layers 7 and 16; the channel partials are visibly asymmetric about $x = 0$ despite a symmetric target. Bottom row: low-rank (RF-LR) with the same hyperparameters; the partials remain symmetric, so the low-rank constraint acts as an implicit regularizer for the target geometry.

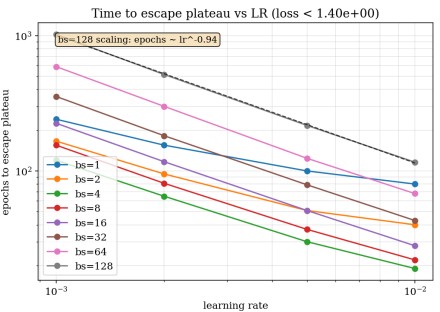 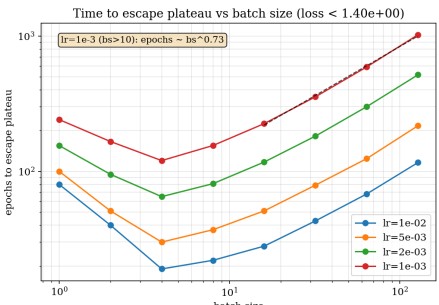

Figure 6: **How quickly the first plateau is left.** Left: epochs to escape the first plateau vs. learning rate; the trend is roughly $1/\text{lr}$. Right: same quantity vs. batch size; small batches (1–4) escape the plateau much sooner than larger ones.

Channel partials evolve over training; Figure 7 shows depth $L = 2$ at the start of training (left) and after 180 epochs (right), with clearer oscillatory structure emerging as the network learns.

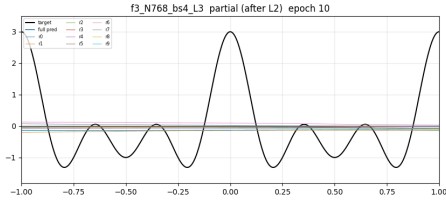 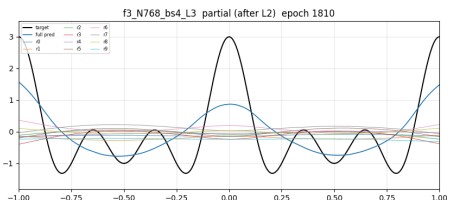

Figure 7: **Channel partials at depth $L = 2$: before and after training.** Left: epoch 0 (initial). Right: epoch 180. Oscillatory structure develops over training as the network captures higher-frequency components.

With Adam, the loss curve can display large symmetric spikes—sudden dips to very low loss followed by a jump back up (Figure 9, Appendix). This pattern is consistent with a landscape that has many deep, narrow valleys: the optimizer can reach near machine-precision loss in one of them and then move out again.

## 4 LIMITATIONS

Our results are limited to 1D regression with cosine targets and the RF-LR architecture; we do not claim that the same feature–landscape relations hold in higher dimensions or for other architectures. The remarks on Adam and on scaling are qualitative, and the oscillatory-complexity numbers come from single runs per configuration. Very deep networks can become unstable. Full configurations and methodology are given in the Appendix.

## 5 CONCLUSION

We have presented a unified picture of feature learning and the loss landscape for RF-LR networks in 1D. The log-ratio criterion and toy model clarify when a channel dominates, and the diagnostics (trajectories, mean minima per channel) support this picture empirically. The loss evolves in a plateau–sharp–plateau fashion, with sharp basins that become accessible when the learning rate is reduced, and observables such as mean minima per channel and the $2L$-spike pattern (we conjecture that layer $L$ has on the order of $2L$ spikes in the partials) link the geometry of the landscape to the way features are learned. Low-rank structure also preserves the symmetry of the target. Within this setting, understanding the loss landscape amounts to understanding when and how the network acquires each frequency.

## REFERENCES

Emmanuel Abbe, Enric Boix-Adsera, and Theodor Misiakiewicz. Sgd learning on neural networks: leap complexity and saddle-to-saddle dynamics, 2023. URL https://arxiv.org/abs/2302.11055.

Lenaic Chizat and Francis Bach. On the global convergence of gradient descent for over-parameterized models using optimal transport, 2018. URL https://arxiv.org/abs/1805.09545.

Arthur Jacot, Franck Gabriel, and Clément Hongler. Neural tangent kernel: Convergence and generalization in neural networks. In *Advances in Neural Information Processing Systems*, pp. 8571–8580, 2018.

Phan-Minh Nguyen and Huy Tuan Pham. A rigorous framework for the mean field limit of multi-layer neural networks, 2023. URL https://arxiv.org/abs/2001.11443.

Huy Tuan Pham and Phan-Minh Nguyen. Global convergence of three-layer neural networks in the mean field regime, 2021. URL https://arxiv.org/abs/2105.05228.

Shijun Zhang, Hongkai Zhao, Yimin Zhong, and Haomin Zhou. Structured and balanced multi-component and multi-layer neural networks, 2025. URL https://arxiv.org/abs/2407.00765.

## A SETUP: ARCHITECTURE, TARGET, DATA, AND TRAINING

### A.1 ARCHITECTURE AND LOW-RANK STRUCTURE

The **low-rank random feature (RF-LR)** network has input/output dimension 1 and $L$ hidden blocks. Each block is: (i) linear rank→width, (ii) ReLU, (iii) linear width→rank, with *rank $r \ll$ width $n$* (tall-skinny bottleneck). Factorized view: weight matrix $W = L R^\top$, $r \ll n$; first-layer features $L^0$ and mixing matrices $L^{(\ell)}$ are **frozen**; only channel weights $w_\ell$ (width→rank linears) are trained. With **fixWb = True**: first linear per block frozen. Main configs: $W = 1024$, $R = 10$, $L \in \{2, 3\}$; $\mu$-parameterization; SGD, momentum 0, gradient clipping 1.0. Channel partial functions $f_k$: scalar outputs per channel; evolution in Appendix B.

### A.2 TARGET, DATA, AND TRAINING

**Target:** $y(x) = \sum_{k=1}^{\text{factor}} \cos(2\pi k x)$ on $x \in [-1, 1]$. **Data:** uniform grid of $N_{\text{train}}$ points; main figures: $N_{\text{train}} = 768$, batch size 8. Loss: MSE. **Training:** initial LR $10^{-2}$; adaptive stagnation (LR halved when loss does not improve over 10 epochs, min 20 between reductions; stop if LR $< 10^{-6}$). Checkpoints: before and 10 epochs after each LR divide. Max 10,000 epochs. Full config naming and sweeps: Appendix C.

### A.3 SYMMETRY AND LOG-RATIO FIGURE SETUPS

Symmetry (Figure 5): 8 layers, $n = 1024$, $r = 20$, target $(f_1, f_2) = (144, 48)$, $N = 4k$, Adam, batch 100. Log-ratio diagnostics: 3-layer, $n = 1024$, $r = 15$, target $\cos(8\pi x)$, $N = 5000$; divergence/convergence: lr $2\times10^{-3}$ bs 4 vs. lr $2\times10^{-4}$ bs 4. Oscillatory complexity: $L = 6$, $W = 128$, $R = 5$; mean minima on layer-$\ell$ partials above $\varepsilon = 10^{-4}$.

## B TWO-SIDED STEP TOY MODEL AND LOG-RATIO CRITERION

### B.1 SETUP AND CLOSED-FORM DYNAMICS

Consider the *two-sided step*: two support points $x_0 = -\delta$, $x_1 = +\delta$ ($\delta > 0$) with $y(x_0) = +A$, $y(x_1) = -A$. For this finitely supported data, define $d_0(t) = d_L((x_0, y(x_0)); W(t))$, $d_1(t) =$

$d_L((x_1, y(x_1)); W(t))$, $B_{k,1}(t) = B_k(t; x_0)$, $B_{k,2}(t) = B_k(t; x_1)$, and

$$\Gamma_{k,1}(t) = \int_0^t \xi_1(s)\, d_0(s)\, B_{k,1}(s)\, ds, \qquad \Gamma_{k,2}(t) = \int_0^t \xi_1(s)\, d_1(s)\, B_{k,2}(s)\, ds.$$

Then $f_k$ takes the form

$$f_k(t, x) = f_k(0, x) - \Gamma_{k,1}(t)\, K_{\mu_0}(x, x_0) - \Gamma_{k,2}(t)\, K_{\mu_0}(x, x_1). \tag{1}$$

The spike shape is determined by $K_{\mu_0}$ (NNGP, Jacot et al. (2018)); all learning dynamics reduce to the scalar coefficients $\Gamma_{k,1}(t)$, $\Gamma_{k,2}(t)$. The kernel satisfies $K_{\mu_0}(x_p, x_p) = K_0 > 0$ for $p \in \{0, 1\}$ (same by symmetry). The off-diagonal $K_{\mu_0}(x_0, x_1) = K_{\mu_0}(-\delta, +\delta)$ is positive and fastly decaying in $\delta$: $0 < K_{\mu_0}(x_0, x_1) \leq \psi(\delta)$ with $\psi(\delta) \to 0$ rapidly as $\delta \to \infty$. Thus the cross-term is small for separated points and, being positive, reinforces the leading local term in the log-ratio dynamics.

The full evolution has two terms (local plus non-local). At $x_0$ and $x_1$:

$$\partial_t f_k(t, x_0) = -\xi_1(t)\, K_{\mu_0}(x_0, x_0)\, d_0(t)\, B_{k,1}(t) + E_0(t), \quad \partial_t f_k(t, x_1) = -\xi_1(t)\, K_{\mu_0}(x_1, x_1)\, d_1(t)\, B_{k,2}(t) + E_1(t), \tag{2}$$

where the non-local remainders satisfy $|E_p(t)| \leq C'\, \psi(\delta)$ with $\psi(\delta)$ fastly decaying in $\delta$.

### B.2 STATEMENT: LOG-RATIO CRITERION AND THEOREM

**Definition B.1** (Log-ratio criterion). At $x_0$, define $R_{12}(t, x_0) = \log \frac{|f_1(t, x_0)|}{|f_2(t, x_0)|}$.

**Theorem B.2** (Two-sided step: log-ratio growth at $x_0$ (conditional)). ***Setting:*** *The two-sided step with $r = 2$ channels and a 3-layer RF-LR; the dynamics are equation 1 and the full evolution equation 2. The result is **conditional**: if the three conditions in (Hypothesis) hold on an interval $I \subseteq [0, \infty)$, then the stated conclusion holds on $I$.*

***Hypothesis (at $x_0$, for all $t \in I$):*** *(i) $-d_0(t) \geq 0$; (ii) $B_{1,1}(t)$ has the same sign as $f_1(t, x_0)$; (iii) there exists $\rho_0 \in [0, 1)$ such that $|B_{2,1}(t)| \leq \rho_0 \frac{|f_2(t, x_0)|}{|f_1(t, x_0)|} |B_{1,1}(t)|$.*

***Conclusion:*** *For $t \in I$,*

$$\partial_t R_{12}(t, x_0) = (1 - \rho_0)\, \xi_1(t)\, K_{\mu_0}(x_0, x_0)\, (-d_0(t))\, \frac{|B_{1,1}(t)|}{|f_1(t, x_0)|} + \varepsilon_0(t),$$

*with $|\varepsilon_0(t)| \leq C''\, \psi(\delta)$ for $\psi(\delta)$ fastly decaying in $\delta$ (from $E_p$). Hence $\partial_t R_{12}(t, x_0) \geq 0$ whenever the leading term dominates $|\varepsilon_0|$; since $K_{\mu_0}(x_0, x_1) > 0$, the off-diagonal coupling contributes with a favorable sign and reinforces the leading term. Strict dominance of channel 1 at $x_0$ cannot be lost on $I$ and is amplified whenever $|B_{1,1}|$ is not too small. The result is symmetrical at $x_1$ (channel 2 dominates there).*

**Remark B.3** (On the hypothesis). The theorem does *not* assert that (i)–(iii) hold for the canonical two-sided step ($y = \pm A$ at $x = \pm \delta$) from generic initial conditions; it only establishes that *when* they hold on $I$, the conclusion follows. Verifying (i)–(iii) from the ODEs for specific initial conditions and $A, \delta$ is outside the scope of this result. When (i)–(iii) hold in practice is discussed below.

### B.3 PROOF OF THEOREM B.2

We prove the theorem at $x_0$; the argument at $x_1$ is symmetrical.

**At $x_0$.** By the log-ratio derivative identity at $x_0$,

$$\partial_t R_{12}(t, x_0) = \frac{\text{sign}(f_1)\, \partial_t f_1}{|f_1|} - \frac{\text{sign}(f_2)\, \partial_t f_2}{|f_2|}.$$

Insert the full evolution equation 2: $\partial_t f_k(t, x_0) = -\xi_1\, K_{\mu_0}(x_0, x_0)\, d_0\, B_{k,1} + E_0(t)$ with $|E_0(t)| \leq C'\, \psi(\delta)$ for $\psi(\delta)$ fastly decaying in $\delta$. Then

$$\partial_t R_{12}(t, x_0) = \xi_1(t)\, K_{\mu_0}(x_0, x_0)\, (-d_0(t)) \Big( \frac{\text{sign}(f_1) B_{1,1}}{|f_1|} - \frac{\text{sign}(f_2) B_{2,1}}{|f_2|} \Big) + \varepsilon_0(t), \qquad |\varepsilon_0(t)| \leq C''\, \psi(\delta),$$

where $\varepsilon_0(t) = E_0(t)\left(\frac{\mathrm{sign}(f_1)}{|f_1|} - \frac{\mathrm{sign}(f_2)}{|f_2|}\right)$ inherits the bound from $E_0$. Use $-d_0 \geq 0$. The sign condition gives $\mathrm{sign}(f_1)B_{1,1} = |B_{1,1}|$; $-\mathrm{sign}(f_2)B_{2,1} \geq -|B_{2,1}|$. The dominance inequality at $x_0$ is $|B_{2,1}| \leq \rho_0 \frac{|f_2|}{|f_1|} |B_{1,1}|$, hence

$$\frac{|B_{1,1}|}{|f_1|} - \frac{\mathrm{sign}(f_2)B_{2,1}}{|f_2|} \geq \frac{|B_{1,1}|}{|f_1|} - \frac{|B_{2,1}|}{|f_2|} \geq (1-\rho_0)\frac{|B_{1,1}|}{|f_1|},$$

so $\partial_t R_{12}(t,x_0) = (1-\rho_0)\,\xi_1\,K_{\mu_0}(x_0,x_0)\,(-d_0)\,\frac{|B_{1,1}|}{|f_1|} + \varepsilon_0(t)$. The leading term is $\geq 0$; since $K_{\mu_0}(x_0,x_1) > 0$ and fastly decaying, the off-diagonal coupling reinforces the leading term while $|\varepsilon_0| \leq C''\,\psi(\delta)$ is negligible for $\delta$ large. Thus $\partial_t R_{12}(t,x_0) \geq 0$ whenever the leading term dominates $|\varepsilon_0|$.

**At $x_1$.** The same argument applies by symmetry (indices $1 \leftrightarrow 2$, $x_0 \leftrightarrow x_1$): channel 2 dominates at $x_1$ and $\partial_t R_{21}(t,x_1) \geq 0$.

**Conclusion.** The log-ratio $R_{12}$ at $x_0$ is non-decreasing on $I$; strict dominance of channel 1 at $x_0$ cannot be lost and is amplified whenever $|B_{1,1}|$ is not too small. By symmetry, channel 2 dominates at $x_1$.

### B.4  On the hypothesis: when (i)–(iii) hold in practice

The hypothesis of Theorem B.2 is conditional: (i) $-d_0(t) \geq 0$, (ii) $B_{1,1}(t)$ has the same sign as $f_1(t,x_0)$, and (iii) $|B_{2,1}(t)| \leq \rho_0 \frac{|f_2(t,x_0)|}{|f_1(t,x_0)|} |B_{1,1}(t)|$ for some $\rho_0 \in [0,1)$. In practice, these conditions are observed to hold in standard training setups. Under *Xavier initialization* (or similar scale-corrected schemes) for the frozen feature and mixing matrices, and *sub-Gaussian initialization* for the trainable weights $w_1$, $w_2$, the initial $f_k$ and backprop signals $B_{k,p}$ are well-balanced across channels. The gradient-flow dynamics then tend to amplify small asymmetries: once one channel leads at $x_0$, (ii) and (iii) are maintained because the dominant channel receives a larger $B_k$ and thus a larger $\partial_t f_k$, while the weaker channel's backprop stays proportionally smaller. Condition (i) holds when the network under-predicts at $x_0$; for squared loss $d_L \propto \hat{y} - y$, $-d_0 \geq 0$ corresponds to $\hat{y}(x_0) \leq y(x_0) = +A$, which is typical before convergence. Thus, although the theorem does not prove (i)–(iii) from first principles, they are consistent with and typically observed under standard initialization.

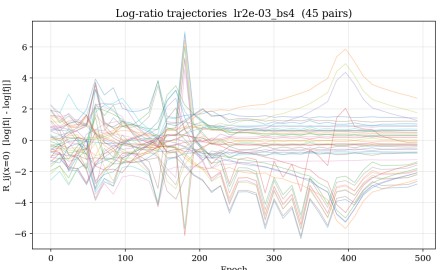
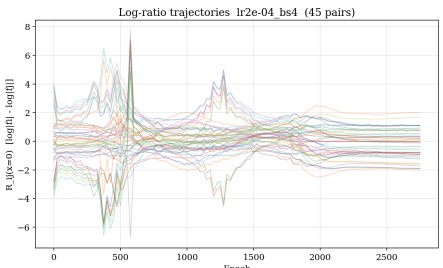

Figure 8: **Specialization vs. collapse as a function of LR and batch.** Left: at higher LR ($2\times10^{-3}$, batch 4) the log-ratio trajectories diverge by end of training and channels do not settle into clear roles. Right: at lower LR ($2\times10^{-4}$, batch 4) the trajectories converge to a stable spread, with channels maintaining distinct contributions. The left panel exhibits a scaling exponent near 1 as expected; the right panel gives $\approx 0.75$, which may relate to $(1+0.5)/2$ from martingale and hole distribution in the landscape (speculative).

## C  Full experiment methodology and configs (exhaustive)

### C.1  Target and architecture (all experiments)

Target: $y(x) = \sum_{k=1}^{\mathrm{factor}} \cos(2\pi k x)$ on $x \in [-1,1]$. Low-rank RF network: input/output dim 1; hidden rank $R$, width $W$; depth $L$ blocks (linear rank$\to$width, ReLU, linear width$\to$rank). fixWb = True. $\mu$-parameterization; SGD momentum 0; gradient clipping max norm 1.0.

## C.2 CONFIG NAMING

A config is identified by **factor**, $N$ (train size), **bs** (batch size), and $L$ (depth). We write this as `f{factor}_N{n}_bs{bs}_L{L}`: e.g. `f3_N768_bs8_L2` means factor $= 3$, $N = 768$, batch size $= 8$, depth $L = 2$; `f5_N1280_bs4_L3` means factor $= 5$, $N = 1280$, bs $= 4$, $L = 3$.

## C.3 MAIN FIGURES (LOSS, FIT, PARTIALS)

Factor 3, $N_{\text{train}} = 768$, batch 8, depths $L = 2, 3$: configs `f3_N768_bs8_L2` and `f3_N768_bs8_L3`. Architecture: width 1024, rank 10. Training: up to 10,000 epochs; initial lr 0.01, divided by 2 on stagnation (window 10 epochs, min 20 epochs between reductions); stop if lr $< 10^{-6}$. Checkpoints: state before and 10 epochs after each LR divide.

## C.4 BASELINE SWEEP (COS FUNCTIONS): DEFINITION AND RANKS

We run baseline sweeps for **rank 5**, **rank 10**, and **rank 20**. Target: cos functions $\sum_{k=1}^{\text{factor}} \cos(2\pi kx)$. **Sweep**: factors 1,2,3,4,5; $N = \text{base} \times \text{factor}$ with base $\in \{16, 32, 64, 128, 256\}$; batch sizes 1, 2, 4, 8, 16 (only bs $\leq N$); depths $L \in 1..2\times\text{factor}$; up to 10k epochs; adaptive LR (halved on plateau; stop if lr $< 10^{-6}$). Config name: `f{factor}_N{n}_bs{bs}_L{L}` (e.g. `f3_N768_bs4_L3` = factor 3, $N = 768$, bs 4, $L = 3$).

## C.5 RANK 5: EXHAUSTIVE COUNTS AND TABLES

**Global (rank 5):** total runs 750; worked (final test error $< 0.01$) 76; failed (test $\geq 0.5$ or NaN/Inf) 674.

**By factor (rank 5):**

| factor | $N$ range | total | worked | failed | best test err |
|---|---|---|---|---|---|
| 1 | 16–256 | 50 | 34 | 16 | 1.88e-05 |
| 2 | 32–512 | 100 | 27 | 73 | 1.13e-04 |
| 3 | 48–768 | 150 | 12 | 138 | 5.26e-04 |
| 4 | 64–1024 | 200 | 2 | 198 | 1.59e-03 |
| 5 | 80–1280 | 250 | 1 | 249 | 3.26e-03 |

**By factor and batch size (rank 5) — worked / total:**

| factor | bs=1 | bs=2 | bs=4 | bs=8 | bs=16 |
|---|---|---|---|---|---|
| 1 | 6/10 | 8/10 | 7/10 | 7/10 | 6/10 |
| 2 | 6/20 | 7/20 | 7/20 | 4/20 | 3/20 |
| 3 | 3/30 | 4/30 | 3/30 | 2/30 | 0/30 |
| 4 | 0/40 | 1/40 | 1/40 | 0/40 | 0/40 |
| 5 | 0/50 | 0/50 | 1/50 | 0/50 | 0/50 |

For factor 5, rank 5, only bs=4 has a worked config: `f5_N1280_bs4_L3` (factor 5, $N = 1280$, bs 4, $L = 3$).

**By batch size only (rank 5) — aggregate:**

| bs | count | worked (test$< 0.01$) | mean(test_err) | best test err |
|---|---|---|---|---|
| 1 | 150 | 15 (10%) | 1.72e+00 | 9.37e-05 |
| 2 | 150 | 20 (13%) | 1.62e+00 | 1.88e-05 |
| 4 | 150 | 19 (13%) | 1.46e+00 | 5.18e-05 |
| 8 | 150 | 13 (9%) | 1.36e+00 | 2.17e-04 |
| 16 | 150 | 9 (6%) | 1.40e+00 | 3.93e-04 |

**Configs with min_loss $< 2\times10^{-2}$ (rank 5).** Factor 5: `f5_N1280_bs4_L3` (min_loss 2.80e-03, final_test 3.26e-03); `f5_N1280_bs2_L2` (9.48e-03, 1.11e-02). Factor 4: `f4_N1024_bs4_L3` (1.42e-03, 1.59e-03); `f4_N1024_bs2_L3` (5.50e-03, 5.94e-03); `f4_N1024_bs1_L2`, `f4_N1024_bs2_L2`. Parameter

counts: $L=3 \rightarrow 36{,}880$; $L=2 \rightarrow 25{,}611$ (formula $2W + L(2Wr + W + r) + (W + 1)$, $W=1024$, $r=5$). The full list of 76 worked configs (sorted by final test error) is available in the supplementary material.

## C.6 RANK 10 (COS FUNCTIONS) BATCH SUMMARY

| bs | count | worked (test< 0.01) | mean(test_err) | best test err |
|----|-------|---------------------|----------------|---------------|
| 1  | 92    | 17 (18%)            | 1.26e+00       | 2.66e-06      |
| 2  | 85    | 18 (21%)            | 1.11e+00       | 1.74e-05      |
| 4  | 84    | 16 (19%)            | 1.01e+00       | 3.44e-05      |
| 8  | 84    | 13 (15%)            | 9.93e-01       | 2.51e-04      |
| 16 | 84    | 9 (11%)             | 1.02e+00       | 5.18e-04      |

Same trend: larger batch (8, 16) $\rightarrow$ fewer worked runs and worse best test error.

## C.7 PLATEAU-ESCAPE AND LOG-RATIO RUNS

Runs with fixed lr and batch size: train until loss < threshold; record **epochs to escape** and **log-ratio** $R_{i,j} = \log |f_i| - \log |f_j|$ at $x = 0$ over time. Main-figure setting: factor 3, $N = 768$, $L = 3$, rank 10, width 1024; escape threshold 0.1; max 400–2000 epochs. Log-ratio trajectory figures: lr $2 \times 10^{-3}$ bs 4 (divergence) and lr $2 \times 10^{-4}$ bs 4 (convergence). Epochs-to-escape scaling (Figures in main text): sweep over lr and batch size; scaling consistent with $\propto 1/\text{lr}$ and small batch (1–4) escaping effectively.

## C.8 FREQUENCY/LAYER-SCALING

Multi-frequency cosine target; freq multipliers 0.3, 0.6, 1.5, 2, 3, 5, 7, 10; ranks 10, 15, 25; layer counts 3, 5, 8, 12, 16, 24, 40, 56, 80. 58 completed configs. Outcomes: low freq (0.3, 0.6) test error $\sim 10^{-6}$–$10^{-5}$; high freq (1.5+) no convergence; very deep ($L = 56, 80$) blow-up or NaN at higher freq (e.g. freq×5, 7).

## C.9 SYMMETRY COMPARISON

Full-rank MLP (fixWb = False): 8 layers, width 1024, $r = 20$, target $f(x) = \cos(f_1 \pi x^2) - 0.8 \cos(f_2 \pi x^2)$ on $[-1, 1]$ with $(f_1, f_2) = (144, 48)$; $N = 4k$, 10k epochs, Adam lr 0.001, batch 100, StepLR $\gamma = 0.9$ every 100. Channel partials at layers 7 and 16 show asymmetry about $x = 0$. Low-rank (RF-LR) with same hyperparameters preserves symmetry.

## C.10 OSCILLATORY-COMPLEXITY (PARTIAL-FUNCTION MINIMA)

Per-layer partial functions on a uniform grid; count of strict local minima above a positivity threshold per channel; layer statistic = mean minima per channel. Outputs: per-layer and per-group stats; figure mean minima per channel vs. layer (e.g. $L = 6$, $W = 128$, $R = 5$).

# D ADAM TRAINING CURVES (ADDITIONAL)

The large symmetric spikes (down then up) in these Adam loss curves suggest machine-precision training is enabled by a landscape with many deep holes (valleys) that the optimizer can enter and leave. Figure 9 shows the factor 2, $N = 512$, batch 512, depth 1 run; the repeated down-then-up spikes indicate that the optimizer repeatedly enters very deep valleys (very low loss) and then leaves them. Additional factor/depth/config variants from the low-lr Adam table follow.

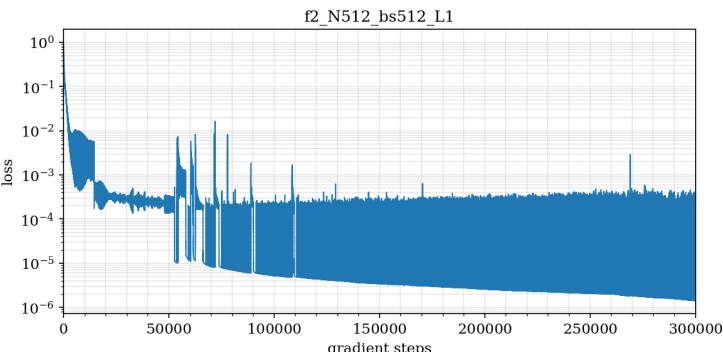

Figure 9: **Adam loss curve showing symmetric spikes.** Factor 2, $N = 512$, batch 512, depth 1. The repeated down-then-up spikes indicate that the optimizer repeatedly enters very deep valleys (very low loss) and then leaves them, supporting the picture of a plateau with many sharp holes.

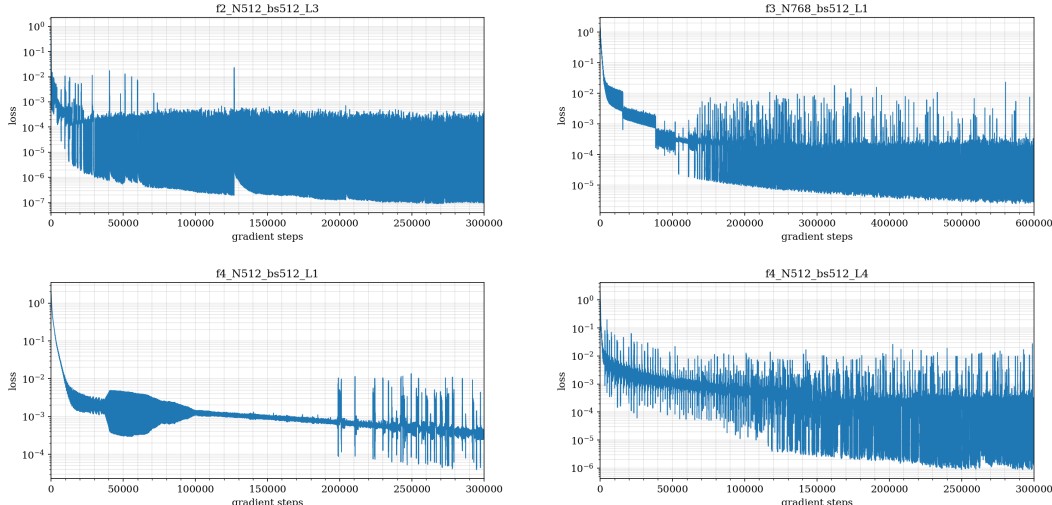

Figure 10: Adam loss curves. **(a)** f2, $N = 512$, bs 512, $L = 3$. **(b)** f3, $N = 768$, bs 512, $L = 1$. **(c)** f4, $N = 512$, bs 512, $L = 1$. **(d)** f4, $N = 512$, bs 512, $L = 4$. Symmetric spikes indicate a landscape with many deep valleys.

