# OpenReview forum: "LOW-RANK NETWORKS LEARN HIGH-FREQUENCY FEATURES ON LOSS LANDSCAPE VALLEYS"
_ICLR.cc/2026/Workshop/Sci4DL — Submitted to Sci4DL 2026_

### Official Review · Reviewer_hbYx · 2026-02-24

**Fit:** 2
**Significance:** 2
**Confidence:** 2

**Summary:**

This paper studies how the geometry of the loss landscape relates to what a low-rank neural network learns. The authors focus on a controlled setting: one-dimensional regression on sums of cosine functions using a low-rank random feature (RF-LR) architecture. They show that training progresses in stages, where the loss remains flat for long periods (plateaus) and then drops sharply. Each sharp drop corresponds to the network learning a new frequency component of the target function. Entering these “valleys” in the loss landscape typically requires reducing the learning rate.

To analyze internal dynamics, the paper introduces two main tools. First, a log-ratio criterion is used to measure when one channel becomes dominant at a specific input location, explaining how channels specialize. Second, the authors measure oscillatory complexity by counting the number of strict local minima in each channel’s partial function; this number increases with depth, suggesting that deeper layers learn higher-frequency structure. They also observe that small batch sizes and lower learning rates help the optimizer escape plateaus more quickly. Finally, the low-rank constraint is shown to preserve symmetry in learned representations, unlike full-rank networks that may break symmetry even when the target is symmetric.

**Strengths:**

A key strength of the paper lies in its conceptual integration of optimization geometry and representational dynamics. Rather than treating feature learning and loss landscape analysis as separate threads, the authors construct observable bridges, log-ratio trajectories, and oscillatory minima counts that connect parameter-space geometry to functional complexity. This gives the work a structural coherence that many empirical landscape studies lack.

Another notable strength is the use of a low-rank architecture as a controlled analytical environment. By freezing feature matrices and restricting training to channel weights, the authors reduce confounding factors while retaining nontrivial nonlinear behavior. This design enables partial theoretical treatment via the two-sided step model and the conditional log-ratio theorem. Although limited in scope, this theoretical component meaningfully complements the empirical findings.

**Suggestions:**

A primary limitation of the work is its narrow experimental scope: all results are derived from one-dimensional cosine regression under a highly structured low-rank RF architecture. Although this restriction is clearly acknowledged, the central mechanistic claims, such as the plateau–frequency correspondence and the conjectured exponential growth of oscillatory structure with depth, may not readily extend to higher-dimensional settings, non-periodic targets, or fully trainable deep networks. In addition, the log-ratio theorem is conditional and does not establish that channel specialization arises generically from random initialization, while some empirical scaling observations remain heuristic.

Given that the paper repeatedly characterizes the landscape in terms of “deep holes,” “narrow valleys,” and saddle-to-saddle transitions, it may be valuable to complement the current dynamical analysis with a more global structural perspective. In particular, recent work such as Evaluating Loss Landscapes from a Topology Perspective (Xie et al.) demonstrates how topological data analysis and persistent homology can quantify connected components, holes, and basin structures of loss sublevel sets. Exploring whether the proposed frequency–valley correspondence admits a topological characterization, for instance, whether frequency acquisition aligns with birth/death events in persistent homology, could strengthen the structural interpretation and potentially improve the generalizability of the geometric picture beyond the controlled 1D regime.

---

### Official Review · Reviewer_jskA · 2026-02-27

**Fit:** 2
**Significance:** 2
**Confidence:** 1

**Summary:**

If I understand it correctly, this paper investigates the relationship between the geometry of the "loss landscape" (during training) and the specific features a neural network "learns" during training. The authors appear to demonstrate that learning does not happen smoothly, but rather through a process where the network traverses flat plateaus until the learning rate is reduced, allowing it to descend into narrow "valleys" that correspond to specific frequency components . Furthermore, the investigations seems to suggest that the depth of the network is critical for complexity, as deeper layers seemingly develop the capacity to represent increasingly oscillatory, high-frequency structures. (Please note that  summary this is based on my own reading as well as "discussing" the paper the paper with Gemini 3 Pro)

**Strengths:**

The paper seems to be a nice analysis of the learning process (features learned over time, how do they map to channels) on a simple network architecture. This seems to have implications on choosing learning rate etc.

**Suggestions:**

To be honest, I did not understand most of the paper, but I am also starting to wonder if I am the target audience. I used Gemini 3 Pro to help me explain the paper, rewrite the introduction in easier terms and explain concepts such as low rank networks to me (I have some background in ML, but low rank networks and random features were not among them. Based on this, it seems the paper is a nice and useful analysis in that it helps to beging to understand the learning process better. Below are a few things that I believe could make the paper more easily readable and accessible to a wider audience, as well as listing points of confusion that may be good to clarify.

* While I understand the need for clear terminology, I found abstract and introduction too heavy in jargon. I think a lot of the concepts could be explained in clearer terms.
* The appendix is already fairly substantial, but I think adding to it could help. Having a brief introduction/discussion of low rank random feature networks and clearly define all the terms used in the paper. Or give references that introduce these concepts in a clear way.
* When reading "loss landscape" I was thinking primarily of the 2D landscapes described by Li, Hao, et al. "Visualizing the loss landscape of neural nets." Advances in neural information processing systems 31 (2018)", i.e. the shape of loss around a trained model. This paper seems to view the shape of the loss during training as "loss landscape". I would recommend to to explicitly define "loss landscape" to disambiguate.
* Similarly, knowing most about convolutional neural networks, I had a wrong concept of "channel" in my mind. Defining this explicitly could help.
* I'd suggest to have a diagram of the used architecture in the appendix and describe the architecture in more detail (and full sentences)

---

### Meta-Review · Area_Chair_KBqx · 2026-02-28

**Recommendation:** Reject

**Metareview:**

This paper studies low-rank random-feature networks on a synthetic regression task. First, as noted by one reviewer, the paper is hard to read and does not give sufficient details to be understood: for instance, the notation $f_i$ in the log-ratio is not defined. Second, the paper does not conduct or discuss experiments on real-world datasets and is therefore unfit for the workshop. I recommend rejection.

---

### Decision · Program_Chairs · 2026-03-02

Reject